

# Technical Note: The impact of industrial activity on the amount of atmospheric $O_2$

Mark O. Battle[1], Raine Raynor[1], Stephen Kesler[2], and Ralph Keeling[3]

[1]Dept. of Physics & Astronomy, Bowdoin College, Brunswick ME 04011-8488 USA
[2]Dept. of Earth and Environmental Sciences, University of Michigan, Ann Arbor, MI 48109 USA
[3]Scripps Institution of Oceanography, UC San Diego, 9500 Gilman Dr. 0244, La Jolla CA 92093 USA

**Correspondence:** Mark O. Battle (mbattle@bowdoin.edu)

**Abstract.**

Concurrent measurements of atmospheric $O_2$ and $CO_2$ amount fractions have been used for decades now to estimate fluxes of carbon to and from the oceans and the land biosphere. The equations used in these estimates explicitly include fossil fuel combustion but largely ignore fluxes of $O_2$ associated with the refining of metals and other industrially important elements.
Here, we quantify the $O_2$ fluxes associated with the processing of iron, aluminum and copper. We also consider the potential impact of sulfur. We find that inclusion of the fluxes due to metals leads to an increased estimate of ocean carbon uptake in the years 2000-2010 of $\left(0.144^{+0.002}_{-0.005}\right)$ Pg a$^{-1}$ with a corresponding decrease in estimated land uptake. A rough estimate of sulfur chemistry during fossil fuel combustion also increases ocean uptake but by a much smaller amount. These corrections are small compared to existing estimates of the fluxes and their uncertainties ($(2.27\pm0.60)$ and $(1.05\pm0.84)$ Pg a$^{-1}$ for ocean and land respectively (Keeling and Manning, 2014)) but should be employed in future analyses.

## 1  Introduction

Since the pioneering work of Keeling and Shertz (1992), measurements of the abundance of atmospheric $O_2$ and $CO_2$ have been used extensively for determining fluxes of carbon to and from the land biosphere and the oceans. The budgets for atmospheric $O_2$ and $CO_2$ can be written as

$$\frac{\mathrm{dn}(O_2)}{\mathrm{dt}} = \alpha_{\mathrm{ff}}\,F_{\mathrm{ff}} + \alpha_B\,F_{\mathrm{land}} + Z_{\mathrm{ocean}} \tag{1}$$

$$\frac{\mathrm{dn}(CO_2)}{\mathrm{dt}} = F_{\mathrm{ff}} + F_{\mathrm{land}} + F_{\mathrm{ocean}} + F_{\mathrm{cem}} \tag{2}$$

where $F_x$ is the flux of $CO_2$ from reservoir $x$ to the atmosphere, $\alpha_{\mathrm{ff}}$ is the global average oxidative ratio of fossil fuels, $\alpha_B$ is the effective average oxidative ratio of the global net land carbon sink and $Z_{\mathrm{ocean}}$ describes the outgassing of $O_2$ from the ocean due to warming. $F_{\mathrm{cem}}$ represents the $CO_2$ source associated with the manufacture of cement.

In this paper we quantify the importance of fluxes omitted from these budget equations: the $O_2$ fluxes associated with the production of iron, aluminum and copper during their extraction from raw materials. These fluxes have been ignored in $O_2$-based calculations of the carbon budget, but as other terms in the budget equations have become increasingly well-constrained,





we consider the size and signficance of a $Z_\mathrm{metals}$ term in Eq. 1. We also show that $O_2$ and $CO_2$ fluxes associated with sulfur are already included conceptually (and approximately) in these budget equations.

We focus on iron, aluminum, copper, and sulfur for two reasons: They are used in great quantities by humanity, and their oxidative states change between extraction and end use. Consequently, they are the heretofore neglected species with the
5   greatest likelihood of influencing the abundance of atmospheric $O_2$.

In the sections that follow, we treat each species in turn. We briefly describe the path from raw material to refined product and then estimate the quantities of product. From these, we calculate fluxes of $O_2$ to/from the atmosphere. In all cases, we focus exclusively on the raw materials that end up as product and ignore fluxes of $O_2$ associated with oxidation or reduction of raw materials that are extracted but *not* refined, such as mine tailings. We conclude by comparing the fluxes from these
10   industrial materials to other terms and uncertainties in the budget.

## 2   Iron

Iron is an abundant and accessible element, enormously useful in both pure form and alloyed with various other metals as steel. In 2021, world production of iron ore contained about $1.6 \times 10^{12}\,\mathrm{kg}$ (Tuck, 2022a) of iron, most of which was used to manufacture steel. The ore had a value of roughly $260 billion USD prior to refining (Statista, 2022c). Iron production was
15   more than 25 times as great (by mass) than the next most heavily produced metal: aluminum (Bray, 2020a). With this ubiquity comes the potential for significant fluxes of $O_2$.

With iron, as with our other species, we determine these fluxes by comparing the oxidative states of the raw and processed materials and scaling by the amount of product. Note that in our accounting, we are *not* concerned with fluxes related to the energy required for production. These are captured in the $F_\mathrm{ff}$ term in equations 1 and 2.

20   ### 2.1   Production Chemistry

Iron and steel are produced primarily from the important iron-bearing minerals hematite ($Fe_2O_3$) and magnetite ($Fe_3O_4$).

At present, hematite is turned into iron by reacting the mineral with carbon monoxide in the presence of heat. This strips the oxygen off the iron and turns the CO into $CO_2$. Although this process typically occurs in three steps, the net reaction is

$$6\,C + 3\,O_2 + 2\,Fe_2O_3 \rightarrow 4\,Fe + 6\,CO_2$$

25   The carbon in this process typically arrives as coke (derived from coal) which is then partially combusted to form CO.

As written, this reaction shows that 3 moles of $O_2$ will be required for every 4 moles of iron produced. However, the central question is *where* the $O_2$ (that ends up in $CO_2$) comes from. Of the 12 oxygen atoms released in $CO_2$, 6 of them came from the hematite, effectively yielding a source of 3 moles of $O_2$ to the atmosphere.

The "$O_2$ source" nature of the process becomes clearer if we recogize that the $F_\mathrm{ff}$ formulation of Eq. 1 assumes the oxidant
30   for fossil fuels is exclusively atmospheric $O_2$. If there is a non-atmospheric oxidant, the equation needs a correction term. In particular, in the reductive refining of metals, the ore itself oxidizes the carbon in the fossil fuel, yielding reduced metal and





$CO_2$ as the products with little or no atmospheric $O_2$ required. Thus, Eq. 1 overestimates $O_2$ loss (relative to $CO_2$ production) whenever oxidized metals are refined using fossil fuel carbon.

As a specific example, consider hematite reduction:

$$2\,Fe_2O_3 \rightarrow 4\,Fe + 3\,O_2$$

5   This immediately shows that producing 4 moles of reduced Fe will release 3 moles of $O_2$. There is an alternative method for hematite processing, but because the end product is again fully reduced iron, the Fe:$O_2$ ratio remains 4:3.

Magnetite is processed slightly differently, with $Fe_3O_4$ and CO reacting to yield FeO and $CO_2$. The FeO is further reduced with CO to yield pure iron with the following net reaction:

$$4\,C + 2\,O_2 + Fe_3O_4 \rightarrow 3\,Fe + 4\,CO_2$$

10   Not surprisingly, the Fe:$O_2$ ratio is 3:2.

## 2.2 Inventory

We use data from the US Geological Survey Mineral Yearbooks (Tuck, 2020a, b) to characterize the global annual production of iron ore from 1990 through 2018. Values are given in Table 1, including both hematite and magnetite. We convert mass of ore to moles of hematite and magnetite by assuming that 80 % of the ore is hematite (Tuck, 2019), but as shown below our

results are relatively insensitive to this assumption. Using the Fe:$O_2$ molar ratios of 4:3 and 3:2 respectively yields the source of atmospheric $O_2$ associated with pig iron production presented in Table 1. A time series of these values is shown in Figure 1.

The details of the fate of iron after it is mined are complicated, with various steps of concentration and refining yielding pig iron and ultimately steel. There are losses along the way, producing substantial quantities of slag and other byproducts. Nonetheless, effectively 100 % of the oxidized iron extracted from the earth is reduced (Tuck, 2021), enabling the calculation

of the atmospheric $O_2$ source given above.

In practice, pig iron contains small amounts of C, S, Si, P and Mn; impurities that are removed as the pig iron is refined into steel. The dominant impurity is carbon, but since it enters the iron from coking coal during smelting its oxidation has been correctly captured by $F_{ff}$ in Eq. 1. Sulfur impurities are also almost entirely introduced by the coking coal, so their oxidation will be described by the treatment of sulfur in fossil fuels (see Sec. 5). Silicon is present in the ore in an oxidized state and

remains so during smelting ($SiO_2 + CaO \rightarrow CaSiO_3$). Phosorus and manganese are more complicated, as they are present in mineral ore in an oxidized state, are reduced early in processing, and are then re-oxidized at a later step. Thus, just as with silicon, phosphorus and manganese have no net $O_2$ flux associated with processing.

## 3 Aluminum

Aluminum, the third most abundant element in the earth's crust, is a relative newcomer to human exploitation, first isolated in

1825 and widely used only after 1886 (Kesler and Simon, 2015). The demand for aluminum has grown steadily since then, due





to its strength, light weight, and corrosion-resistance, leading to a commercial value second only to iron among metals. Global production in 2021 was $6.8 \times 10^{10}$ kg (Bray, 2022) with a value of \$170 billion USD (Statista, 2022a). Large as this is, the production is only $8\,\%$ (by mole) of that of iron, implying a relatively modest impact on atmospheric oxygen.

### 3.1 Production Chemistry

Essentially all aluminum extracted for human use comes from bauxite, an ore containing various hydrated aluminum oxides, along with iron oxides and various other impurities. The primary oxides are diaspore and böhmite (both $AlO(OH)$), and gibbsite ($Al(OH)_3$). About $85\,\%$ of bauxite that is mined is converted to alumina ($Al_2O_3$) (Merrill, 2022) using the Bayer process of leaching and calcination. The abundances (both relative and total) of $AlO(OH)$ and $Al(OH)_3$ vary substantially from one deposit of bauxite to the next, but these two species have the same oxygen yields during refining. The remaining $15\,\%$
of the bauxite is used for a variety of purposes but the aluminum in it is not reduced (Bray, 2021).

 Of the alumina produced, the great majority ( $88\,\%$ in 2017) (Bray, 2020b, a) is reduced to pure aluminum using the Hall-Heroult process. The remaining alumina is not further reduced, and is used instead in chemicals, abrasives and other products (Bray, 2021).

 While the Bayer process (converting bauxite to alumina) is quite complicated, the net effect is a partial reduction of the
hydrated oxides. Most importantly for our purposes, the liberated oxygen ends up as water. Thus, ignoring links between the oxygen and water cycles, the production of alumina has no impact on atmospheric $O_2$.

 In contrast, the reduction of alumina to aluminum using Hall-Heroult involves an electrochemical transfer of oxygen from the alumina to carbon from graphite anodes. The consequent release of $CO_2$ is, once again, effectively a source of atmospheric $O_2$ since there is oxidation of carbon with no impact on atmospheric $O_2$ levels. Since the reduction of alumina can be represented
as

$$2\,Al_2O_3 \rightarrow 4\,Al + 3\,O_2$$

the ratio of aluminum to $O_2$ is 4:3.

### 3.2 Inventory

As with iron, we use data from the US Geological Survey Mineral Yearbooks (Bray, 2020b, a) for global annual production of
pure aluminum from 1990 through 2017. Values are given in Table 2.

 Since there is no $O_2$ flux associated with the Bayer process, we only need to account for the refining of alumina to aluminum. Applying the 4:3 ratio given above yields the values shown in Table 2 as well as Figure 1.

## 4 Copper

Copper was one of the first metals used by humans and played a pivotal role in the development of civilization, first in
pure form and later alloyed with tin as bronze. In modern society it remains essential, serving crucial functions in electrical





power generation and electronics, plumbing, and marine applications. In 2021, roughly $2.1 \times 10^{10}$ kg of copper was mined worldwide (Flanagan, 2022). This had a value of roughly \$196 billion USD (Statista, 2022b). While these figures clearly show the global importance of copper, the production is only $1.1\%$ of iron (by mole), suggesting a smaller influence on atmospheric oxygen than either iron or aluminum. The quantity of copper recycled and reclaimed is substantial (of order $5 \times 10^9$ kg) but its

processing does not generate $O_2$ fluxes, so we ignore it in the work that follows.

## 4.1 Production Chemistry

The chemistry of copper production is more complicated than that of iron and aluminum since some copper-bearing minerals are sulfides while others are oxides. The dominant sulfides and their approximate fractional molar abundances are chalcopyrite ( $CuFeS_2$ 70 %), chalcocite ($Cu_2S$ 15 %) and bornite ($Cu_5FeS_4$ 10 %). The oxides are primarily malachite ($Cu_2(CO_3)(OH)_2$),

atacamite ($Cu_2Cl(OH)_3$), brochantite ($Cu_4SO_4(OH)_6$) and chrysocolla ($(Cu, Al)_2H_2Si_2O_5(OH)_4 \cdot nH_2O$), all roughly equally abundant (Sillitoe, 2021).

      Sulfides account for about $85\%$ of copper production (Schlesinger et al., 2011). Because the sulfides contain no oxygen, they cannot be a source to the atmosphere. Instead, the reduced sulfur and iron liberated during processing are eventually oxidized, functioning as a sink of atmospheric $O_2$.

Elemental copper in chalcopyrite and bornite is extracted by concentration and smelting, followed by electrolytic refinement. The iron largely ends up in slag as FeO or $Fe_2O_3$ and the sulfur is emitted as gaseous $SO_2$. Copper in chalcocite is refined by leaching, followed by solvent extraction and electrowinning (SX-EW). This yields elemental sulfur. Regardless of the mineral type, the extraction method, or the intermediate oxidation state, the sulfur that is liberated nearly always ends up as sulfuric acid with three of the oxygen atoms in the $H_2SO_4$ coming from gaseous oxygen while the last comes from liquid

water (Schlesinger et al., 2011). In the calculations that follow, as with aluminum we ignore any linkage between atmospheric $O_2$ and the water cycle. For chalcocite this implies a sink of $O_2$ in the molar ratio 4:3 (Cu:$O_2$). For chalcopyrite and bornite, the ratios are 8:29 and 40:53 respectively, assuming equal amounts of ferric and ferrous oxide in the slag.

      An additional complication for the sulfides is the risk of double-counting $O_2$ fluxes associated with the oxidation of sulfur, since we develop a separate $O_2$ budget for this element. We choose to include the $O_2$ sink associated with copper sulfides in

the copper budget rather than the sulfur budget, allowing us to limit the sulfur budget calculations to the reduced sulfur present in fossil fuels.

      For the copper oxides, elemental copper is produced using the same methods employed for chalcocite, described above (leaching and SX-EW). The chemistry of the leaching is complicated by the variety of oxides to be considered and by the aqueous nature of the process. For example, when a single molecule of malachite is leached (by sulfuric acid), its five oxygens

end up in one $CO_2$ and three water molecules. Rather than try to account for the many possiblities, we simply ignore the small source of atmospheric $O_2$ originating from processing of copper oxides, recognizing that oxides only account for 15% of copper production which is itself a small fraction of iron production.





## 4.2 Inventory

Again, we use data from the US Geological Survey Mineral Yearbooks (Flanagan, 2021) for global annual production of copper ore from 1990 through 2017. Values are given in Table 3.

We assume $85\%$ sulfides, with the relative mineral species proportions (scaled to $100\%$ abundance) and $Cu:O_2$ ratios given above. Ignoring any $O_2$ flux associated with the processing of oxides, we obtain the sink of atmospheric $O_2$ given in Table 3 and shown in Figure 1. We recognize that ignoring oxides leads to a slightly exaggerated atmospheric $O_2$ sink and discuss this further in Section 6.

## 5 Sulfur

Sulfur is widely used as an industrial raw material, and is essential in the world's fertilizer and manufacturing sectors. The great majority of sulfur ends up as sulfuric acid, the most abundant inorganic chemical produced in the United States (Apodaca, 2022). The total world production of sulfur in 2017 was 80.2 million metric tonnes (Mt), slightly more than aluminum on a molar basis.

Significantly, only about $8\%$ (as of 2017) of the world demand for sulfur is met by "discretionary" production, in which sulfur or iron sulfides (usually pyrite) are mined from discrete deposits (Apodaca, 2022). The remaining $92\%$ of the world demand is met by "nondiscretionary " production: sulfur, sulfur dioxide and sulfuric acid recovered as byproducts of other process. The byproducts have significant commercial value and their abundance has led to almost complete cessation of Frasch-process sulfur mining (Ober et al., 2016).

The dominance of nondiscretionary production is a relatively recent phenomenon. Production of $H_2SO_4$ from fossil fuels began in the mid-1970s as regulation limited $SO_2$ emissions. Global emissions of $SO_2$ peaked at $130\,Mt$ in the late 1980s and have fallen steadily since (Ober et al., 2016). The captured sulfur has almost completely replaced discretionary production. Globally, non-discretionary production now exceeds demand, driving down sulfur prices and leading to growing challenges for storage and disposal (Ober et al., 2016).

Thanks to this transition to non-discretionary production, sulfur is in a different category than iron, aluminum and copper: Current production of sulfur is almost entirely associated with fossil fuel production. This has two consequences. First, the sulfur in fossil fuels is in a reduced state and is subsequently oxidized, serving as a *sink* of atmospheric $O_2$, much like the copper sulfides discussed above.

Second, we can largely incorporate sulfur directly into the $O_2$ budget through careful handling of fossil fuel combustion with no need for separate $O_2$ fluxes in Eq. 1. This was essentially done by Keeling (1988) when he used $\alpha_{ff}$ to characterize the fluxes of $O_2$ associated with the combuston of non-carbon species present in fossil fuels. At the time of Keeling's work, discretionary sulfur production was substantial. In the years since, it has become negligible, so Keeling's formulation has the potential to completely account for the sulfur-driven sink of atmospheric $O_2$. In brief, the value of $\alpha_{ff}$ is a consumption-weighted average of $\alpha_{gas}$, $\alpha_{liquid}$ and $\alpha_{solid}$. These fuel-specific oxidative ratios are built on the abundance of C, H, S & N in



the fuels, assuming (among other things) the sulfur ends up as $H_2SO_4$. Thus, in Eq. 1, $\alpha_{ff}$ captures the $O_2$ flux associated with all non-discretionary sulfur production.

In practice, this approach to budgeting has complications that are beyond the scope of this paper. For example, the chemistry of flue gas desulfurization changes both the $O_2$ and $CO_2$ fluxes somewhat. It is also possible that the amount of sulfur in the coal, oil and gas reserves currently being exploited is different from those in use nearly 40 years ago. Discretionary production of sulfur is not utterly negligible at 8% of the total. Natural gas (essentially $CH_4$) contains no sulfur, but is very often found with $H_2S$. The latter is captured pre-combustion and converted to $H_2SO_4$, consuming $O_2$. Additionally, correctly characterizing uncertainties is particularly challenging.

While we choose to defer a rigorous calculation, we can make a very rough estimate of the currently-neglected $O_2$ sink associated with sulfur production. In 2017 world production of sulfur included $17\,\mathrm{Tg}$ from metallurgy, $49\,\mathrm{Tg}$ from liquid and gaseous fuels, $140\,\mathrm{Tg}$ from solid fuels and $13\,\mathrm{Tg}$ from sulfides and other discretionary production (Apodaca, 2022; Gilfillan and Marland, 2021; Keeling, 1988). If we assume each sulfur atom sinks 1.5 $O_2$ molecules for solid fuels and 2.0 in every other case, this implies a sulfur-driven $O_2$ sink of $11.5\,\mathrm{Tmol}$. However, the $\alpha_{ff}$ formulation of Keeling et al. (1998) takes nearly all of the $O_2$ sink associated with the sulfur in liquid and solid fuels into account. This leaves a sink of $2.4\,\mathrm{Tmol}$. We discuss the significance of this residual term in Sec. 6.2 below.

# 6 Results, Uncertainties and Discussion

## 6.1 Metals

Collectively, Tables 1, 2 and 3 show that the production of useful iron, aluminum and copper from mined minerals resulted in a net annual flux of $O_2$ to the atmosphere of between 21 and $22\,\mathrm{Tmol\,a^{-1}}$ in 2017, depending on assumptions about the mix of iron species refined into metals. Averaged over the period 2000-2010, the range is 11.6-12.2 with a most likely value of $12.0\,\mathrm{Tmol\,a^{-1}}$. This flux ($Z_{\mathrm{metals}}$) should be added into Eq. 1. Like all of the other terms in Eqs. 1 and 2, $Z_{\mathrm{metals}}$ varies over time. To facilitate incorporation of $Z_{\mathrm{metals}}$ in future $O_2$-based carbon budgets, the total flux shown in Fig. 1 is also given (with uncertainties) as Table 4. The uncertainty in these fluxes is difficult to rigorously quantify due to the assumptions involved in their estimation.

The largest uncertainty of which we are aware comes from the hematite/magnetite mix from which iron is refined. While an 80:20 $(\mathrm{mol\,mol^{-1}})$ mix is our best estimate, we only know with certainty that a majority of iron comes from hematite but *some* comes from magnetite. Thus, we consider 95:5 and 50:50 mixes as extreme cases around our 80:20 central value. These result in an asymmetric range of values for the estimated $O_2$ flux from iron. In 2018, the flux was $(21.37\,^{+0.34}_{-0.68})\,\mathrm{Tmol\,O_2}$.

Luckily, uncertain aspects of the processing of iron (*e.g.* losses to slag and other byproducts as iron ore becomes pig iron, or losses during transport) are irrelevant, as essentially $100\,\%$ of the iron in the ore is reduced, regardless of whether it ends up in its intended form (Tuck, 2021).

We take the total production figures for iron, aluminum and copper from the USGS, which does not formally calculate uncertainties. Although presented by convention with three significant figures, the global values for iron production are likely





good to $0.75\,\%$ or better (Tuck, 2022c). Global copper production is significantly less certain (roughly $5\,\%$ for solvent-extraction/electrowinning and $10\,\%$ for primary smelting (Flanagan, 2022)). We assume similar uncertainty for aluminum ($10\,\%$). For 2018, these imply uncertainties in $O_2$ fluxes of $0.15, 0.03$ and $0.18\,\mathrm{Tmol}\,O_2$, respectively.

Our treatment of copper has several additional uncertainties and biases. We assume that iron present in copper sulfides is
equally likely to be found in FeO and $Fe_2O_3$ states, but the true distribution is unknown. We assume an 85:15 split on copper sulfides and oxides, but do not know the uncertainty in that split. Lastly, we completely ignore the roughly $15\,\%$ that are oxides because of the diversity of oxide minerals and the complexity of the processing. This omission will bias $Z_{\mathrm{metals}}$ low.

While our ignorance of uncertainties in copper production is unfortunate, it has little impact on the uncertainty in $Z_{\mathrm{metals}}$ simply because production of copper is so much smaller than production of iron. Even if the $O_2$ fluxes from copper carry a
$25\,\%$ uncertainty (an arbitrary, but very conservative number, allowing for the production uncertainty given above and $23\,\%$ uncertainty from other sources), the 2018 $O_2$ flux from copper will be known to $\pm0.09\,\mathrm{Tmol}$.

Uncertainty in $O_2$ fluxes from aluminum are driven entirely by uncertainties in production so the value of $0.18\,\mathrm{Tmol}\,O_2$ (for 2018) quoted above is the whole story.

It is worth noting that we have limited the scope of our study to iron, aluminum and copper. The oxide ores of copper and
other metals make negligible contributions to the oxygen budget simply because we refine less of them. However, sulfide ores of zinc, nickel, lead and molybdenum are more widely exploited. In total, the amount of these metals extracted from sulfides is comparable, by mole, to copper sulfides. We also ignore the passive oxidation of iron pyrite ($FeS_2$) in mine tailings, which may lead to $O_2$ fluxes comparable to the other sulfides. Our neglect of these various sulfides biases our value of $Z_{\mathrm{metals}}$ high (since they are $O_2$ sinks). Conveniently, this sulfide-driven bias is comparable in size and in the opposite sense of the neglected-oxides
bias described above.

While the list of quantified uncertainties given above is neither rigorous nor exhaustive, it is dominated by a single term: the uncertainty in the hematite-magnetite mix. In 2018, this accounts for 0.51 of the total uncertainty in $Z_{\mathrm{metals}}$ of $\pm0.57\mathrm{Tmol}\,a^{-1}$. Other years are similar. Furthermore, the uncertainty in $Z_{\mathrm{metals}}$ is small compared to other uncertainties in the $O_2$ budget equations.

$Z_{\mathrm{metals}}$ should be compared to the various fluxes already in Eq. 1. During the 2000-2010 period, Keeling and Manning (2014) report annual $O_2$ fluxes of $F_{\mathrm{ff}} = (934\pm56)\,\mathrm{Tmol}\,a^{-1}$ and $Z_{\mathrm{ocean}} = (44\pm45)\,\mathrm{Tmol}\,a^{-1}$. The inferred net land flux is $F_{\mathrm{land}} = (96\pm77)\,\mathrm{Tmol}\,a^{-1}$. $Z_{\mathrm{metals}}$ ($11.6-12.2\,\mathrm{Tmol}\,a^{-1}$) is much smaller than $F_{\mathrm{ff}}$ and very likely smaller than $Z_{\mathrm{ocean}}$. It is also substantially smaller than the uncertainty in $F_{\mathrm{land}}$ (arising from uncertainties in $F_{\mathrm{ff}}$, $Z_{\mathrm{ocean}}$ and observed $\mathrm{dnO_2/dt}$ (Keeling and Manning, 2014)).

Nonetheless, including $Z_{\mathrm{metals}}$ in Eq. 1 gives a more accurate model of the $O_2$ budget and should be done for $O_2$-budgeting work going forward. $Z_{\mathrm{metals}}$ becomes still more significant when Atmospheric Potential Oxygen ($APO \equiv O_2 + \alpha_B CO_2$) (Stephens et al., 1998) is the focus, rather than carbon fluxes. Since $Z_{\mathrm{metals}}$ is solely an $O_2$ flux, it becomes relatively large when other paired fluxes partially cancel.

Adding $Z_{\mathrm{metals}}$ to Eq. 1 and repeating the APO-based analysis of Keeling and Manning (2014) for the years 2000-2010,
the oceanic carbon sink increases by $\left(0.144^{+0.002}_{-0.005}\right)\,\mathrm{Pg}\,a^{-1}$ and the land sink decreases equivalently. If we take the average



of our extreme values and use half the range as the uncertainty, the ocean and land sinks become $(2.86 \pm 0.60)$ and $(0.92 \pm 0.84)\,\mathrm{Pg\,a^{-1}}$.

Looking decades or more ahead, our framework for calculating $Z_{\mathrm{metals}}$ will likely need adjustment. Efforts are underway to refine iron-bearing minerals at lower temperatures using hydrogen-rich reductants, releasing some oxygen as $H_2O$ rather than

$CO_2$. Bauxite will inevitably be replaced by other aluminum-bearing oxides and silicates. For copper and other base metals, the proportion of sulfide ores will increase as we mine deeper deposits that have not been weathered.

Moving from production to consumption, the reduced state of iron has a finite lifetime in our oxygen-rich atmosphere, leading to a sink of atmospheric $O_2$. This is very likely a small effect for two reasons. First, the absolute amount of rusting is small. About $85\,\%$ of steel materials are recycled well before they are fully oxidized (Tuck, 2022b) and much of the steel in

exterior use is either formulated as stainless, or is coated periodically to prevent oxidation. Second, iron production in recent decades has grown rapidly (Table 1) causing a large disequilibrium between production and decay that renders the rust sink negligible. Should iron production stabilize or slow in the future, this correction to $Z_{\mathrm{metals}}$ will be worthy of further attention.

## 6.2  Sulfur

As mentioned in Sec.5, we have chosen to defer a rigorous calculation of the $O_2$ sink resulting from sulfur oxidation. Much

of this sink is already included in Eq. 1 due to the formulation of $\alpha_{\mathrm{ff}}$, but there remains a sink of order $2.4\,\mathrm{Tmol\,a^{-1}}$ that is currently missing.

This value (approximate as it is) should be compared to the $12\,\mathrm{Tmol\,a^{-1}}$ sink represented by $Z_{\mathrm{metals}}$. If we were to introduce $Z_{\mathrm{sulfur}} = 2.4\,\mathrm{Tmol\,a^{-1}}$ into Eq. 1, it would increase the ocean carbon sink for 2000-2010 by $0.03\,\mathrm{Pg\,a^{-1}}$ and decrease the land carbon sink by the same amount. This is clearly a very small correction to substantially uncertain terms ($(2.86 \pm 0.60)$

and $(0.92 \pm 0.84)\,\mathrm{Pg\,a^{-1}}$ for ocean and land, respectively).

Given the small size of the correction and the complexity of a rigorous treatment, we choose to leave a full analysis for later, simply acknowledging that all $O_2$-based ocean (land) carbon sink estimates are biased very slightly low (high) even after improving their fidelity with $Z_{\mathrm{metals}}$.

*Author contributions.* MB conducted research and wrote the manuscript, RR conducted research, RK posed the problem, SK consulted on

the science and all authors reviewed and revised the manuscript.

*Competing interests.* The authors have no competing interests.



*Acknowledgements.* We thank Cristopher Tuck, Joyce Ober, Lee Bray, Daniel Flanagan, Richard Sillitoe, Lori Apodaca, Vincent Camobreco, Christopher Cassar, Gregg Marland and Dennis Gilfillan for helpful conversations and correspondence. Raine Raynor's work was supported by an E.O. LaCasce Jr. Physics Fellowship at Bowdoin College.



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



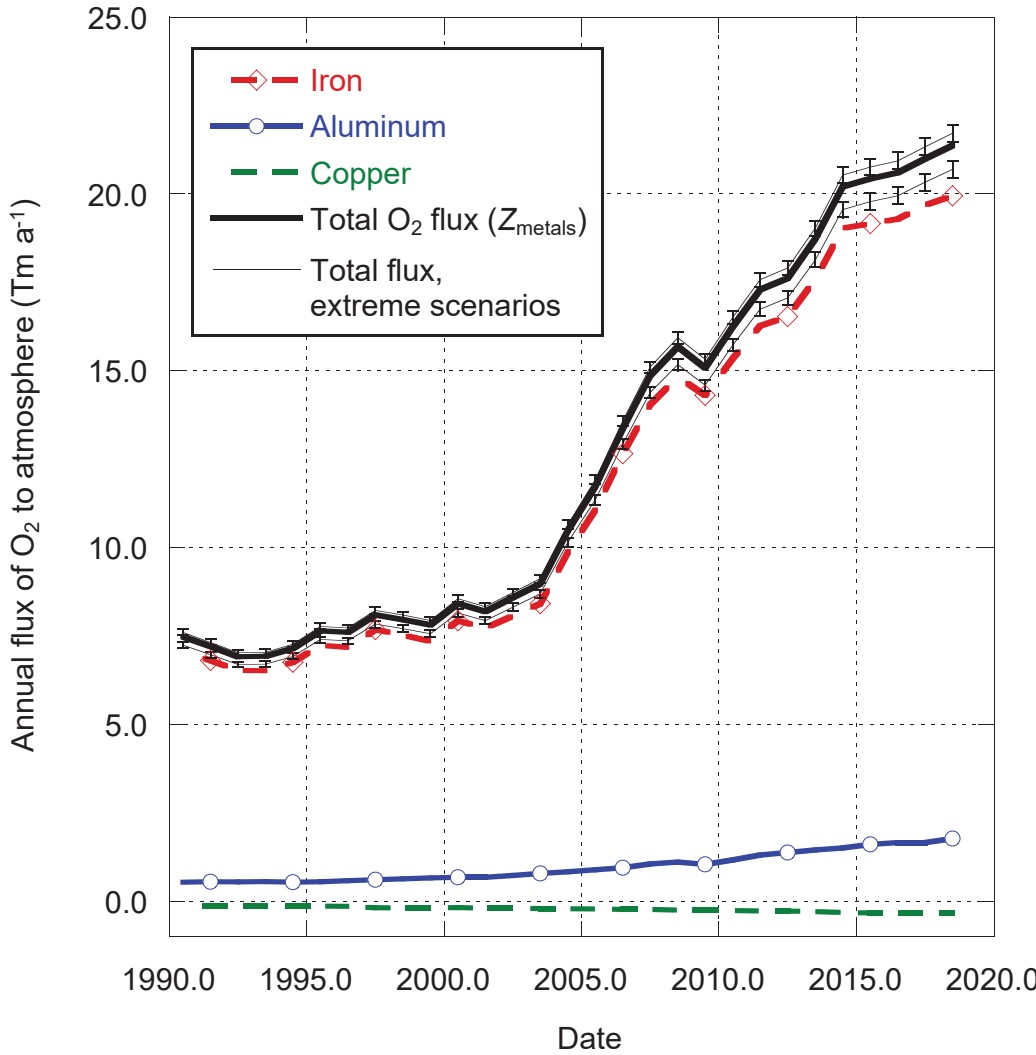

**Figure 1.** Fluxes of $O_2$ to the atmosphere associated with production of various metals ($10^{12}$ mol a$^{-1}$). Negative values indicate a flux *from* the atmosphere. We assume iron is derived from $80\%$ hematite and $20\%$ magnetite (by mole). We assume copper is derived exclusively from sulfides. The heavy black line is the best estimate of total flux to the atmosphere from all sources (assuming an 80/20 mix of hematite and magnetite). The thin black lines show extreme cases (assuming $95\%$ hematite and $50\%$ hematite). Error bars on the extreme cases show the additional uncertainty ($1\sigma$) *not* originating from the unknown hematite/magnetite mix. Values are taken from Table 4, assigning the total annual value to the mid-year point and interpolating linearly.




| Year | World production of reduced iron (Pg) | O₂ source (Tm) assuming 80% hematite | O₂ source (Tm) assuming 50% hematite | O₂ source (Tm) assuming 95% hematite |
|---|---|---|---|---|
| 1990 | 0.54 | 7.09 | 6.85 | 7.21 |
| 1991 | 0.52 | 6.80 | 6.57 | 6.92 |
| 1992 | 0.50 | 6.52 | 6.30 | 6.63 |
| 1993 | 0.50 | 6.52 | 6.30 | 6.63 |
| 1994 | 0.51 | 6.76 | 6.53 | 6.87 |
| 1995 | 0.55 | 7.24 | 6.99 | 7.36 |
| 1996 | 0.55 | 7.18 | 6.93 | 7.30 |
| 1997 | 0.59 | 7.68 | 7.42 | 7.81 |
| 1998 | 0.57 | 7.52 | 7.27 | 7.65 |
| 1999 | 0.56 | 7.35 | 7.10 | 7.47 |
| 2000 | 0.60 | 7.93 | 7.66 | 8.06 |
| 2001 | 0.59 | 7.72 | 7.46 | 7.85 |
| 2002 | 0.61 | 8.07 | 7.79 | 8.21 |
| 2003 | 0.64 | 8.41 | 8.12 | 8.55 |
| 2004 | 0.75 | 9.87 | 9.53 | 10.04 |
| 2005 | 0.84 | 11.06 | 10.69 | 11.25 |
| 2006 | 0.96 | 12.65 | 12.22 | 12.87 |
| 2007 | 1.07 | 14.04 | 13.56 | 14.28 |
| 2008 | 1.13 | 14.83 | 14.33 | 15.08 |
| 2009 | 1.09 | 14.31 | 13.82 | 14.55 |
| 2010 | 1.17 | 15.36 | 14.83 | 15.62 |
| 2011 | 1.24 | 16.27 | 15.72 | 16.55 |
| 2012 | 1.26 | 16.54 | 15.97 | 16.82 |
| 2013 | 1.34 | 17.59 | 16.99 | 17.89 |
| 2014 | 1.45 | 19.03 | 18.38 | 19.36 |
| 2015 | 1.46 | 19.16 | 18.51 | 19.49 |
| 2016 | 1.47 | 19.29 | 18.64 | 19.62 |
| 2017 | 1.50 | 19.69 | 19.02 | 20.02 |
| 2018 | 1.52 | 19.95 | 19.27 | 20.29 |

**Table 1.** World production of iron and the associated flux of $O_2$ to the atmosphere. While the most likely mix of iron-bearing minerals is 80 % hematite and 20 % magnetite (by mole), we include other (extreme) scenarios as a sensitivity study. Data from Tuck (2020a).





| Year | World production of reduced aluminum (Tg) | $O_2$ source (Tm) from reduction of alumina |
|------|------|------|
| 1990 | 19.30 | 0.54 |
| 1991 | 19.70 | 0.55 |
| 1992 | 19.50 | 0.54 |
| 1993 | 19.80 | 0.55 |
| 1994 | 19.20 | 0.53 |
| 1995 | 19.70 | 0.55 |
| 1996 | 20.80 | 0.58 |
| 1997 | 21.70 | 0.60 |
| 1998 | 22.60 | 0.63 |
| 1999 | 23.60 | 0.66 |
| 2000 | 24.30 | 0.68 |
| 2001 | 24.30 | 0.68 |
| 2002 | 26.10 | 0.73 |
| 2003 | 28.00 | 0.78 |
| 2004 | 29.90 | 0.83 |
| 2005 | 31.90 | 0.89 |
| 2006 | 33.90 | 0.94 |
| 2007 | 37.90 | 1.05 |
| 2008 | 39.70 | 1.10 |
| 2009 | 37.20 | 1.03 |
| 2010 | 41.80 | 1.16 |
| 2011 | 46.80 | 1.30 |
| 2012 | 49.30 | 1.37 |
| 2013 | 52.10 | 1.45 |
| 2014 | 54.10 | 1.50 |
| 2015 | 57.80 | 1.61 |
| 2016 | 59.50 | 1.65 |
| 2017 | 59.50 | 1.65 |
| 2018 | 63.60 | 1.77 |
| 2019 | 63.20 | 1.76 |

**Table 2.** World production of aluminum and the associated flux of $O_2$ to the atmosphere. Data from Bray (2020b, a).



| Year | World production of reduced copper (Tg) | O$_2$ sink (Tm) sulfides only |
|---|---|---|
| 1990 | 8.62 | 0.15 |
| 1991 | 8.65 | 0.15 |
| 1992 | 8.73 | 0.15 |
| 1993 | 8.42 | 0.14 |
| 1994 | 8.43 | 0.14 |
| 1995 | 8.49 | 0.14 |
| 1996 | 9.05 | 0.15 |
| 1997 | 11.14 | 0.19 |
| 1998 | 11.40 | 0.19 |
| 1999 | 11.61 | 0.20 |
| 2000 | 11.02 | 0.19 |
| 2001 | 12.11 | 0.20 |
| 2002 | 12.19 | 0.21 |
| 2003 | 12.59 | 0.21 |
| 2004 | 12.81 | 0.22 |
| 2005 | 13.19 | 0.22 |
| 2006 | 13.73 | 0.23 |
| 2007 | 13.88 | 0.23 |
| 2008 | 15.03 | 0.25 |
| 2009 | 15.38 | 0.26 |
| 2010 | 15.85 | 0.27 |
| 2011 | 16.40 | 0.28 |
| 2012 | 16.88 | 0.29 |
| 2013 | 17.74 | 0.30 |
| 2014 | 19.20 | 0.32 |
| 2015 | 19.63 | 0.33 |
| 2016 | 19.96 | 0.34 |
| 2017 | 19.95 | 0.34 |
| 2018 | 20.27 | 0.34 |

**Table 3.** World production of copper and the associated flux of O$_2$ to the atmosphere (in this case, a *removal* of O$_2$ from the atmosphere). We ignore copper produced from oxide minerals and assume the following mix of sulfide species: 70 % chalcopyrite, 15 % chalcocite and 10 % bornite (Sillitoe, 2021) with O$_2$:Cu molar ratios of 29:8, 3:4 and 53:40, respectively. Production data from Flanagan (2021).



| Year | Total $O_2$ sink (Tmol) |
|---|---|
| 1990 | $7.48\,^{+0.15}_{-0.26}$ |
| 1991 | $7.20\,^{+0.14}_{-0.25}$ |
| 1992 | $6.92\,^{+0.14}_{-0.24}$ |
| 1993 | $6.93\,^{+0.14}_{-0.24}$ |
| 1994 | $7.15\,^{+0.14}_{-0.24}$ |
| 1995 | $7.64\,^{+0.15}_{-0.26}$ |
| 1996 | $7.60\,^{+0.15}_{-0.26}$ |
| 1997 | $8.09\,^{+0.16}_{-0.28}$ |
| 1998 | $7.96\,^{+0.16}_{-0.27}$ |
| 1999 | $7.81\,^{+0.16}_{-0.27}$ |
| 2000 | $8.42\,^{+0.17}_{-0.29}$ |
| 2001 | $8.19\,^{+0.17}_{-0.28}$ |
| 2002 | $8.59\,^{+0.17}_{-0.30}$ |
| 2003 | $8.98\,^{+0.18}_{-0.31}$ |
| 2004 | $10.48\,^{+0.21}_{-0.36}$ |
| 2005 | $11.73\,^{+0.23}_{-0.40}$ |
| 2006 | $13.36\,^{+0.26}_{-0.46}$ |
| 2007 | $14.86\,^{+0.29}_{-0.50}$ |
| 2008 | $15.68\,^{+0.30}_{-0.53}$ |
| 2009 | $15.08\,^{+0.29}_{-0.51}$ |
| 2010 | $16.25\,^{+0.32}_{-0.55}$ |
| 2011 | $17.30\,^{+0.34}_{-0.59}$ |
| 2012 | $17.62\,^{+0.34}_{-0.60}$ |
| 2013 | $18.74\,^{+0.37}_{-0.64}$ |
| 2014 | $20.21\,^{+0.39}_{-0.69}$ |
| 2015 | $20.44\,^{+0.40}_{-0.69}$ |
| 2016 | $20.61\,^{+0.40}_{-0.70}$ |
| 2017 | $21.00\,^{+0.41}_{-0.71}$ |
| 2018 | $21.37\,^{+0.42}_{-0.72}$ |

**Table 4.** Net oxygen sink associated with all metals for each calendar year (*i.e.* the annually integrated value of $Z_{\mathrm{metals}}$). Units are $10^{12}$ moles (Tmol). Upper and lower uncertainties are the quadrature sum of the variations arising from the hematite/magnetite scenarios and all other errors.