# Peer review of "Technical Note: The impact of industrial activity on the amount of atmospheric $O_2$"

_Atmospheric Chemistry and Physics, 2022_

## Author Comment (AC1)

**Referee #1:** Thank you for your comments.  Regarding your specific suggestions:

a) We have added "sulfur dioxide" where $SO_2$ first appears.
b) You are correct that it should be Keeling 1988.  Thanks for catching this mistake!
c) We have consulted with the ACP editor.  Given this is a European journal, we have switched to "aluminium".

**Referee #2:** Thank you for your careful reading of the paper.

Comments:

We will attempt to clear up the confusion about the involvement of atmospheric $O_2$ in the production of metals.

Consider first a simplified version of fossil fuel combustion:  nothing but idealized coal.  Assuming it is pure carbon, every mole that is combusted (in the presence of atmospheric air) will release a mole of $CO_2$.  The oxygen in that $CO_2$ was removed from the atmosphere, decreasing atmospheric $O_2$ by 1 mole.  The release of $CO_2$ is expressed in Eq.2.  The consumption of oxygen shows up in Eq.1, and in this coal-only case, alpha$_{ff}$ would be 1.0.

Next, consider the refining of hematite into iron.  Fundamentally, this is just oxidized iron that we're trying to reduce.  It is important to understand that the reaction on p3-line4 is *not* saying that hematite reduction will happen spontaneously.  It definitely won't.  Instead, this is just a shorthand version of the reaction on p2-line24, hiding the role of carbon so that the iron:oxygen stoichiometry is explicit.

The only way to effect the reduction of hematite is to provide a reactant that has a higher affinity for the bound oxygen than the iron.  On an industrial scale, the reactant of choice is CO, derived from incomplete oxidation of coal. Follow the process, starting with coal:  6 carbon atoms grab 3 $O_2$ from the atmosphere to make 6 CO.  In the presence of heat and hematite, this CO then strips 6 more oxygen atoms from 2 hematite units, yielding 4 bare iron atoms and 6 $CO_2$.   Just to emphasize – we have released 6 $CO_2$, while removing 3 $O_2$ from the atmosphere.

Since Eq.1 is built around the assumption that every $CO_2$ released to the atmosphere consumed one molecule of *atmospheric* $O_2$, the equation needs a correction term (Z_metals); effectively a source of $O_2$ to compensate for the $O_2$ that was erroneously subtracted from the atmosphere.

This is what we were trying to communicate on p2-line21 through p3-line2.  We have substantially changed the introduction in an effort to make this particular line of reasoning clearer.  We have also added the qualifier "effective" to the word "flux" when it is used to describe the reduction of oxide ores.

Returning to your second comment (about the structure of eqs 1&2):  You are correct that we could have different $\alpha_{xx}$ for each of the different metal productions, eliminating $Z_{metals}$ entirely.  However, this would also require separate $F_{xx}$ terms for each of the metallic minerals in which carbon plays a role, broken out from $F_{ff}$.  This would be a much more difficult accounting exercise and would yield a much messier set of equations.   This is why we have chosen a formulation with a single correction term.

Minor comments (Line & page numbers refer to the draft on which you commented, not the revised draft) :

1) (p1-L20) We have added a brief mention of the (partial)  independence of $O_2$ and $CO_2$ to the introduction, as you suggest.
2) (p2-L25) Yes, $\alpha$ for coal + $O_2$ + hematite -> iron + $CO_2$ would indeed be 0.5, but as discussed above, we are deliberately avoiding that approach to quantifying fluxes.
3) (p30-L21) Done.
4) (p4-L21) We prefer not to include the aluminum equations for two reasons:  First, the reactions that reduce the various aluminum oxides are complicated.  The Bayer process is a multistep protocol involving several reagents (see the Wikipedia page for an overview).  Second, in sum, these many reactions result in no fluxes of $CO_2$ or $O_2$ to the atmosphere.  Instead, the oxygen that is removed from the aluminum ends up as water.
5) (p5-L22) As you recognize, there are numerous copper-bearing mineral forms and their smelting reactions are significantly more complicated than those of either iron or aluminum.  We feel that adding them would be a distraction.  In terms of consistency across species, we could take two approaches:  Give every reaction that contributes to the production of each of the metals, or simply use the initial and final oxidation states to infer an $O_2$ flux, real or effective.  For clarity and simplicity, we have chosen the latter and try to make it explicit in all three cases (Fe, Cu, Al).
6) (p7-L14) We added some clarification of the origin of the "2.4" value.
7) (p8-L25) We appreciate the wish for a reference point for the size of these fluxes. Since this paragraph deals with oxygen fluxes (rather than carbon fluxes), we have added the requested quantities to the last paragraph on the page.
8) (p8-L35) Throughout the manuscript, we have now changed "Pg" to "PgC" as requested, but we are told that $a^{-1}$ is the IUPAC standard for yearly values (rather than $yr^{-1}$).
9) (Tables 1,2,3)  Thank you for catching this mistake.  We have corrected these units (to Tmol) in the tables and also in Figure 1.
10) (Table and figure captions)  You're correct: These are fluxes only in the sense in which we choose to add them to Eq.1.  We have added "effective" (as noted above) to make this more explicit.

**Referee #3:** Thank you for your suggestions (Line & page numbers refer to the draft on which you commented, not the revised draft) :

1) (p1-L8-10, p8-L25-33) You are correct that $Z_{metals}$ is quite a bit smaller than the uncertainties in the ocean and land sinks.  However, $Z_{metals}$ represents correction of a systematic error, rather than the random errors that currently limit our knowledge of the sinks.  As such, it should be routinely included in the budget calculations.  As for APO: You are also correct that $Z_{metals}$ has the largest impact on the secular trend calculated by Resplandy *et al.*.  We have added a discussion of this to the body of the paper.  Finally, $Z_{metals}$ also leads to a small systematic correction to the heat uptake estimated by Resplandy *et al.*.  We have added a discussion of this as well.
2) (p9-L17 & table 4):  You are correct.  There were several places where we were simply careless and transposed "source" and "sink".  We have fixed all such mistakes.
3) (throughout)  Done to the best of our ability.  In some cases the irregular spacing appears to be an artifact of LaTex that is beyond our control.  Presumably this will be fixed when the article is officially typeset.